# YAP1 Regulates the YAP1/AR/PSA Axis through Autophagy in Castration-Resistant Prostate Cancer and Mediates T-Cell Immune and Inflammatory Cytokine Infiltration

**DOI:** 10.3390/biomedicines12030661

**Published:** 2024-03-15

**Authors:** Youzhi Wang, Ning Wu, Junbo Li, Diansheng Zhou, Jiaming Liang, Qian Cao, Zhaokai Guan, Yangyang Xu, Ning Jiang

**Affiliations:** 1Department of Urology, Tianjin Institute of Urology, The Second Hospital of Tianjin Medical University, Tianjin Medical University, Tianjin 300211, China; 2State Key Laboratory of Female Fertility Promotion, Center for Reproductive Medicine, Department of Obstetrics and Gynecology, Peking University Third Hospital, Beijing 100191, China

**Keywords:** YAP1, AR, PSA, autophagy, CRPC, immune infiltration, inflammatory cytokines

## Abstract

The emergence of castration-resistant prostate cancer (CRPC) following androgen deprivation therapy (ADT) is associated with increased malignancy and limited treatment options. This study aims to investigate potential connections between immune cell infiltration and inflammatory cytokines with the YAP1/AR/PSA axis by exploring their interactions with autophagy. Our research reveals heightened levels of Yes-associated protein 1 (YAP1) expression in CRPC tissues compared with tissues from androgen-dependent prostate cancer (ADPC) and benign prostate hyperplasia (BPH). Additionally, a correlation was observed between YAP1 and PSA expressions in CRPC tissues, suggesting that YAP1 may exert a regulatory influence on PSA expression within CRPC. Enhanced YAP1 expression in C4-2 cells resulted in the upregulation of androgen receptor (AR) nuclear translocation and intracellular prostate-specific antigen (PSA) levels. Conversely, the suppression of YAP1 led to a decrease in PSA expression, suggesting that YAP1 may positively regulate the PSA in castration-resistant prostate cancer (CRPC) by facilitating AR nuclear import. The modulation of the autophagy activity exerts a significant impact on the expression levels of YAP1, the AR, and the PSA. Moreover, recent advancements in immunity and inflammation studies present promising avenues for potential therapies targeting prostate cancer (PC).

## 1. Introduction

Prostate cancer (PC) is the second most prevalent type of malignancy and ranks fifth in terms of male mortality attributed to cancer-related complications [1]. ADT serves as the primary treatment for newly diagnosed or recurrent advanced PC, relying on the signaling pathway mediated by the AR throughout the disease’s progression [2]. Although ADT initially triggers tumor remission, it eventually leads to resistance, culminating in CRPC, a relapse marked by heightened aggression and frequent metastasis [3]. Patients with CRPC experience unfavorable clinical outcomes, with median survival ranging from 9 to 30 months [4]. The advancement of PC to the castration-resistant stage frequently leads to metastasis, notably with a high prevalence of bone metastasis. Additionally, there is a growing inclination toward employing triplet therapy, which includes androgen-receptor-signaling inhibitors, docetaxel, and androgen deprivation therapy, for managing CRPC. Nevertheless, it is important to acknowledge that these treatment approaches may exhibit cross-resistance. Enhanced patient survival has been observed with current therapeutic approaches targeting the androgen receptor pathway, such as enzalutamide and abiraterone. However, their efficacy diminishes over time [5]. Nonetheless, emerging evidence suggests that the activation of alternative oncogenic-signaling pathways plays a pivotal role in circumventing dependence on androgens during the progression of CRPC [6,7,8,9].

Initially recognized as a conserved regulator of tissue growth, the Hippo–YAP pathway has been found to govern tumor chemotherapy resistance, progression, and metastasis across different cancers [10]. Previous studies have underscored the pivotal role played by this signaling pathway in the development of CRPC [11]. Furthermore, there is a significant correlation between the nuclear localization of YAP1 in primary tumors and tumor recurrence following the initial treatment [12]. Further, it is worth exploring whether YAP1 is increased in CRPC compared with primary tumors. Additionally, in laboratory experiments involving androgen-responsive LNCaP cells, the introduction of YAP triggers their transition to an androgen-independent state, thereby facilitating the development of resistance [13].

Autophagy, also known as macroautophagy, is a cellular mechanism responsible for the degradation of large molecules and organelles within cells [8]. Its primary function involves maintaining cellular homeostasis by eliminating persistent and potentially harmful substances. Recent studies have focused on understanding the impact of autophagy on cancer development and its potential as a target for effective cancer treatment [14,15]. Autophagy has demonstrated both promotional and inhibitory effects on the growth and survival of PC cells. Furthermore, it has been observed to regulate the metastatic behavior of these cells through either inhibition or induction [8]. Additionally, autophagy can influence how PC cells respond to chemotherapy and radiotherapy owing to its close association with programmed cell death, also known as apoptosis [16]. Several upstream regulators, including the KLF5 transcription factor and PI3K/AKT/mTOR-signaling pathway, have been identified based on emerging evidence from various research studies [17].

In the present study, we hypothesized the possible regulatory role of autophagy on the YAP1/AR/PSA axis. It is further concluded that YAP1 and its downstream proteins may be related to immunity and inflammation. Finally, it may provide a therapeutic framework for the treatment of CRPC.

## 2. Materials and Methods

### 2.1. Cell Lines and Cell Cultures

AR-negative cells, including PC3 and DU145 cells, were obtained from the University of Texas M.D. Anderson Cancer Center. AR-positive cells, which are androgen-dependent LNCaP cells, were received from Professor Chuan-Xiang Zhang at the University of Rochester in the United States. Androgen-independent C4-2 cells were acquired from the University of Texas M.D. Anderson Cancer Center. All the cell lines (C4-2, LNCaP, PC3, and DU-145) were cultured using a mixture of 1640 medium and 10% FBS at a ratio of 1:9 on aseptic workstations to ensure sterility. The cell cultures were maintained in a sterile incubator at a constant temperature (37 °C) and supplied with 5% CO_2_ under high humidity conditions.

### 2.2. Patients and Tissue Specimens

This study involved individuals with prostate disorders who were admitted to the Urology Department at the Second Affiliated Hospital of Tianjin Medical University. They were randomly selected based on their specific conditions, including 6 cases of BPH, 14 cases of ADPC, and 5 cases of CRPC, and had undergone hormonal therapy for over a year, resulting in testosterone levels reaching castration levels but with inadequate control over PSA levels. All the patients underwent surgical intervention, and the final diagnosis was confirmed by the pathology department at the same hospital. Pathological samples were obtained from the Urology Institute at the hospital, fixed in a solution containing 10% formaldehyde, embedded in wax blocks, and sectioned into paraffin sections measuring 3 μm for immunofluorescence staining.

### 2.3. Western Blot (WB)

Protein extraction from LNCaP, C4-2, PC3, and DU145 cells was performed using PMSF and RIPA buffers. The concentration of different proteins was determined by employing a BCA kit. Subsequently, protein samples were subjected to SDS–polyacrylamide gel electrophoresis on a 10% acrylamide gel followed by transfer to a polyvinylidene fluoride membrane. Afterward, the membrane was blocked with skimmed milk powder at a concentration of 5% and incubated overnight at 4 °C with primary antibodies (GAPDH diluted to a 1:1000 ratio (ab9485), β-actin diluted to a 1:1000 ratio (ab213262), YAP1 diluted to a 1:1000 ratio (ab52771), AR diluted to a 1:1000 ratio (ab209491), and PSA diluted to a 1:1000 ratio (ab76113)). Subsequent washes were carried out twice using PBS before exposing the membrane to anti-mouse IgG/anti-rabbit IgG at room temperature for one hour. Finally, after another round of washing with PBS solution, the protein bands were detected utilizing an automated WB chemiluminescence imaging system.

### 2.4. RNA Extraction, Reverse Transcription, and Quantitative Real-Time PCR (qRT-PCR)

The PC cells and tissues were treated with Trizol reagent (Invitrogen, Waltham, MA, USA) for the total RNA extraction, followed by reverse transcription to cDNA using M-MLV reverse transcriptase (Promega, Madison, WI, USA) according to the manufacturer’s instructions. 

### 2.5. Immunofluorescence Staining

C4-2 cells (2 × 104) were cultured on coverslips in a 24-well plate until they reached approximately 70% confluence. Subsequently, we used PBS to wash the cells and fixed the cells with paraformaldehyde. Next, we blocked the cells with Triton X-100. After that, an anti-YAP1 rabbit antibody was incubated with the cells overnight at a temperature of 4 °C. After this incubation period, the nuclei were stained using DAPI [18,19]. 

### 2.6. MTT Assay

After transfection for 48 h, 2.0 × 103 cells were seeded per well in a 96-well plate and incubated at 37 °C for durations of 24, 48, 72, and 96 h. Subsequently, each well was treated with a 3-(4,5-dimethylthiazol-2-yl)-2,5-diphenyl tetrazolium bromide (MTT) solution for two hours at the same temperature. Formazan crystals were dissolved by adding dimethyl sulfoxide (DMSO) at a volume of 150 μL to each well. Finally, the absorbance at a wavelength of 490 nm was measured using a microplate reader [20].

### 2.7. Clone Formation Assay

We seeded 2.0 × 103 C4-2 cells in each well of a 6-well plate as the initial step. After incubation for 24 h, transfection experiments involving a control, YAP1 WT1, and YAP1 WT2 were performed on the aforementioned cells. Subsequently, the PC cells were cultured for approximately 1–2 weeks before undergoing two rounds of PBS washing to cleanse the plates. Following this, the cells were fixed using paraformaldehyde and subjected to another round of PBS washing to ensure a thorough cleaning. Finally, a crystal violet stain was applied to the cells for half an hour, followed by two additional rounds of PBS washing before allowing them to dry.

### 2.8. Transwell Migration

We separately transfected control, YAP1 WT1, and YAP1 WT2 constructs into cells. Post-transfection, we seeded 2 × 104 cells on a Transwell insert containing 1640 medium (10% FBS) in the upper chamber, while the lower chamber was filled with 1640 medium (10% FBS). The cells were then incubated at 37 °C for 48 h. Following the incubation period, we performed two washes of the chambers using PBS. Subsequently, paraformaldehyde was used to fix the cells, which were then washed twice with PBS. Finally, crystal violet staining was applied to the cells for a duration of 1 h [21].

### 2.9. Tumor Xenograft Mouse Models

Five-week-old immunodeficient nude mice with their thymus removed were used for the experiment. The weights of the mice were 18–25 g. The mouse bedding was sterilized by standard autoclaving, and the mice were fed with sterile water and specialized feed before use. Male mice were injected with 2 × 106 C4-2 cells, suspended in 150 μL of Matrigel and 1640 medium, under the skin of the abdomen in the control, YAP1 WT1, YAP1 WT2 groups. Tumor volume data were collected for at least 2 weeks and measured at the same time every day. Finally, the mice were sacrificed, and weights of the tumors were measured with precision [22]. All the mice were cared for following strict ethical guidelines and were killed by cervical dislocation.

### 2.10. Statistical Analysis

The continuous data were summarized by calculating the mean ± standard deviation for normally distributed datasets and by determining the median along with the interquartile range for datasets that did not follow a normal distribution pattern. For comparing normally distributed datasets, we applied the t-test; whereas, in the case of non-normally distributed datasets, we employed the Mann–Whitney U test instead. In our cross-sectional study design, Spearman correlation analysis was conducted to investigate potential associations among variables of interest. Statistical significance was defined as *p* values of < 0.05 throughout this study’s analyses, which were performed using the GraphPad Prism version 8.0 software package [23].

## 3. Results

### 3.1. Distributions and Expressions of YAP1 and PSA Proteins in Different CRPC Tissues and Cell Lines

Our screening method was developed by combining the expression patterns of YAP1 and the PSA. The YAP1 protein is predominantly localized in the nuclei of cells, exhibiting a higher intensity in CRPC tissue compared with BPH and ADPC tissues. Within specific regions of the ADPC tissue, there is an upregulation of the YAP1 protein expression as the Gleason score increases. Conversely, the PSA primarily localizes to the cytoplasmic region of prostate cells, with a decreased intensity observed in individuals with a low Gleason score. By analyzing both subcellular distribution and expression profiles for YAP1 and the PSA, we can infer that there are limited areas where they coexist within ADPC and BPH tissues, suggesting the lack of a significant correlation. However, in CRPC tissues, cells displaying elevated levels of YAP1 also exhibit increased PSA expression, indicating clear colocalization between these markers (Figure 1A,B). The luciferin reporter gene assay is a detection method utilized for measuring the activity of firefly luciferase by employing luciferin as a substrate. During this process, oxidation occurs to luciferin, and it transforms to oxyluciferin, resulting in the emission of bioluminescence. The emitted bioluminescence can be quantified using fluorescence analyzer equipment. According to the instructions provided by the Promega luciferase test kit, a statistically significant luciferase activity detection value in the treatment group is defined as being at least two times higher than that in the control group. In LNCaP cells, although there was less than a two-fold difference in the logarithmic values of detected signals between the YAP1-overexpression group and control group, there still existed a greater than two-fold difference (*p* < 0.001), indicating statistical significance. Moreover, compared to the serum treatment alone, the addition of 10 nmol/L of YAP1 resulted in a more than two-fold upregulation in the overexpression group compared with the control group (*p* < 0.001), which also demonstrated statistical significance (Figure 1C). Subsequently, we conducted experiments on various cell lines, including both androgen-dependent and non-dependent ones, to validate the expression levels of YAP1 and the PSA under different conditions. DU145 and PC3 cells are AR-negative PC cell lines in which no PSA protein expression was detected, even after transfection with YAP1 cDNA. No significant change in the PSA expression level was observed among LNCaP cells overexpressing YAP1 after the CD serum treatment. However, C4-2 cells exhibited significantly higher levels of PSA expression in the YAP1-overexpression group compared with the control group (Figure 1D). Finally, we transfected C4-2 cells with different fragments of YAP1-overexpression cDNA (WT, S94A, WW, and S369A). After 48 h, RNA extraction was performed followed by reverse transcription to cDNA. The mRNA levels of the PSA were found to be significantly elevated across all the groups with YAP1 overexpression compared to the control group (Figure 1E).

### 3.2. YAP1 Promotes the Progression of CRPC Both In Vivo and In Vitro

The growth pattern of the cells was plotted, and the absorbance of the cell proliferation was evaluated in three groups: the control group, YAP1 WT1 group, and YAP1 WT2 group. The results indicated that cells with elevated levels of YAP1 exhibited a significantly enhanced proliferative capacity compared to the control group (Figure 2A). Subsequently, a lower cell density (2 × 103 cells/chamber) was cultured. After 7–14 days, a noticeable increase in cell proliferation was observed in the high-YAP1-expression group (Figure 2B,C). Recent studies have highlighted the heightened presence of YAP1 in invasive tumor cells and its pivotal role in their proliferative and invasive abilities. The current findings revealed that migration and invasion were amplified in the high-YAP1-expression group when compared with the control group (Figure 2D,E, respectively). C4-2 cells were transfected with either control plasmids or WT plasmids for YAP1 and subsequently subcutaneously implanted beneath the abdominal skin of nude mice. After a duration of 2 weeks, significant differences in weight measurements were observed between the group treated with YAP1 WT plasmids and the control group (Figure 2F–H).

### 3.3. Autophagy Regulates the YAP1/AR/PSA Axis in CRPC

The GEPIA platform possesses the capability to extract information through both single gene and multi-gene analyses, facilitating cross-analysis across diverse cancer types. Moreover, it facilitates the retrieval of multiple gene expressions, encompassing survival analysis and correlation between genes. The AR pathway, positioned upstream of the PSA and playing a pivotal role in the progression of CRPC, was investigated for its correlation with YAP1 using the GEPIA online database [24]. Our analysis revealed a positive association between YAP1 and the AR (Figure 3A). Fluorescence double-staining results demonstrated predominant nuclear expressions of both the AR and YAP1 in PC cell lines, encompassing both CRPC (represented by C4-2 cells) and ADPC (represented by LNCaP cells). Notably, high-power microscopy revealed an evident enhancement in the fluorescence intensity observed in YAP1-overexpressed cells. A comparison was conducted between the control group and the treatment group in terms of changes in the AR expression intensity. It is evident that there exists a positive correlation between the YAP1 expression intensity and AR localization within the nuclei of C4-2 cells, while no significant correlation is observed in LNCaP cells (Figure 3B). Upon silencing the YAP1 gene through siRNA transfection in C4-2 cells, noticeable alterations were observed in both the distribution and location of the AR within the cellular compartments. In the control group, the AR predominantly localized to the nucleus, whereas upon silencing, it mainly resided in the cytoplasm (Figure 3C). This further supports a role for the YAP1 protein in regulating the nuclear entry of the AR. WB experiments confirmed this finding as well; the overexpression of YAP1 resulted in substantial increases in both AR protein expression and its downstream PSA levels (Figure 3D). Similarly, the knockout of YAP1 led to significant reductions in both AR and PSA expressions (Figure 3E). To investigate whether autophagy is involved with respect to modulating the YAP1/AR/PSA axis, we either inhibited this process using 3-MA and siRNA against LC3B or stimulated it with rapamycin. The results aligned with our assumptions: the inhibition of macroautophagy elevated levels of YAP1, while the rapamycin treatment decreased its levels (Figure 3F–H).

### 3.4. Correlations of YAP1 with the Proportion of Tumor-Infiltrating Immune Cells and Inflammatory Cytokines in PC

The efficacy of immunotherapy in treating PC has been suggested to be limited in several studies [25]. However, recent advancements in understanding immune mechanisms and molecular diagnostics have revitalized interest in utilizing immunotherapy as a promising therapeutic option for patients with CRPC. The objective is to stimulate their innate antitumor immunity [26,27]. Cancer immunotherapy aims to enhance the activity of cytotoxic T lymphocytes (CTLs) within tumors, facilitate tumor-specific CTL priming in lymphoid organs, and establish long-lasting antitumor immunity [28,29]. During this priming phase, CD4+ T cells transmit signals to CD8+ T cells through dendritic cells, optimizing both the magnitude and quality of the CTL response. The TIMER2.0 platform integrates six advanced algorithms to provide a more robust assessment of immune infiltration levels in the Cancer Genome Atlas (TCGA) or user-provided tumor profiles. It offers four modules dedicated for investigating connections between immune penetration and genetic or clinical characteristics, as well as four modules designed to explore cancer-related associations within the TCGA cohort [30,31]. Using TIMER2.0 software analysis (http://timer.cistrome.org/) (Figure 4A–F) in our study, we observed positive correlations among the YAP1 expression, AR and PSA activities, and various immune cell infiltrates within the human tumor microenvironment. Notably, these correlations were found to be significant for both CD4+ and CD8+ T cells present in PC. The CIBERSORT algorithm utilizes linear-support-vector regression to decode the expression matrices of distinct immune cell subtypes, providing estimations for the prevalence of immune cells. It furnishes comprehensive data on 22 prevalent immunoinfiltrating cells, encompassing a diverse array of immune cell types and functional states. Next, we utilized the CIBERSORT algorithm to evaluate the immune cell subset’s composition within tumor infiltrates. This analysis, conducted on 276 tumor samples, with a significance level of *p* < 0.05, identified 22 distinct immune cell profiles in PC samples (Figure 5A). These findings highlighted T cells, specifically CD4-memory-resting and CD8 T cells, as the predominant immunoinfiltrating cells associated with PC (Figure 5B). The establishment of durable immune protection against previous pathogens or antigens is attributed to the pivotal role played by CD4 T cells in establishing long-term immunological memory. However, ongoing debates persist regarding the mechanisms underlying the formation and maintenance of these memory CD4 T cells, partly owing to ambiguous interpretations of what constitutes T-cell memory. The current understanding emphasizes a multitude of pathways involved in shaping and sustaining persistent populations of memory T cells. We next classified PC into two distinct groups: normal and PC groups. Notably, T cells and macrophages, particularly regulatory T cells and macrophages M0 and M2, emerged as the predominant types of immunoinfiltrating cells in PC (Figure 5C). In the TCGA cohorts, we utilized the weighted gene co-expression network analysis (WGCNA) methodology to identify and categorize 5000 genes into five gene modules. Importantly, our findings demonstrated robust associations between the brown module and resting activity of CD4 memory T cells. Specifically, YAP1 was identified as a member of the brown module (Figure 5D). The differentiation of effector T cells heavily relies on cytokines released by antigen-presenting cells (APCs). To gain deeper insights into the relationship between YAP1 and APCs in T cells, we conducted a protein–protein interaction (PPI) analysis. Our PPI analysis revealed several connections linking YAP1 with proteins associated with APCs, including SKP1, SMAD7, CDH17, MPP5, and TEAD1. (Figure 6A). By stratifying YAP1 based on its expression levels, we were able to identify two distinct groups and investigate the correlation between YAP1 and immune stimulators. Interestingly, the group characterized by a high expression of YAP1 exhibited elevated levels of various factors related to inflammation, such as IL-10, STAT3, and STAT4 among others (Figure 6B). To further validate these observations, we utilized the GEPIA online database for correlation analysis, which confirmed positive associations between YAP1 and these inflammatory cytokines (Figure 6C).

## 4. Discussion

Improving the survival rate of patients with CRPC remains an urgent issue. Owing to different treatment methods and types of surgery, their subsequent treatment presents a variety of options. The development of a variety of effective and precise molecular target therapies has a good effect on the treatment of CRPC. The activation of the AR is crucial for the continuous growth of CRPC and is characterized by an aberrant reactivation of AR signaling. However, attempts to target androgen synthesis using second-generation AR antagonists and inhibitors to further suppress AR signaling have not yielded a curative outcome. Despite prolonging survival and increasing survival rates, the progression of the disease remains inevitable owing to the highly heterogeneous nature of CRPC tumors, which exhibit diverse clinical outcomes. Additionally, the advancement of CRPC may involve triggering alternative pathways that circumvent or compensate for disrupted signaling. Our research findings indicate that YAP1 plays a pivotal role in driving the progression of CRPC. Immunofluorescence results showed that YAP1 may play a role in CRPC by assisting the AR to enter the nucleus and further regulate the changes in the PSA downstream of the AR. Mechanistically, YAP1 activates autophagy signaling through the YAP1/AR/PSA pathway, thereby promoting the growth of castration-resistant cells. By inhibiting autophagy, we can regulate this process and suppress AR nuclear translocation and expression, ultimately impacting downstream PSA levels. Taken together, these data support a hypothetical mechanism for the conversion from the androgen-dependent to the androgen-independent growth of PC, involving YAP1 upregulation and subsequent binding to the AR, leading to the activation of AR signaling. It may be noted that previous studies have shown that energetic stress due to ADT turns on the autophagic response, expediting the transition of PC cells from androgen dependence to androgen independence. It is suspected that the enhanced autophagy due to AR-signaling inhibition therapy and subsequent autophagy deficiency may upregulate YAP signaling through some unrecognized mechanisms. Importantly, YAP1, the AR, and the PSA were strongly associated with CD4+/CD8+ T cells in PC compared with other cancers (Figure 4 and Figure 5, respectively). This result suggests their potential as targets for immunotherapy against CRPC.

Frequent upregulation and/or nuclear translocation of YAP1 in human cancers suggest(s) persistent activation of the YAP/TAZ pathway [32]. The Hippo pathway is regulated by various biological processes, including mechanotransduction, intercellular contact, and cellular polarity [33]. Furthermore, disruptions in the Hippo pathway can arise from epigenetic modifications and genetic as well as post-transcriptional regulations of components within the signaling cascade [34]. A deeper understanding of the regulatory mechanisms governing the Hippo–YAP pathway may lead to innovative therapeutic targets for PC [35]. Recent findings have indicated that the inhibition of IKBKE triggers the degradation of LATS2, resulting in elevated levels of YAP, which drive tumorigenesis in PC. Targeting IKBKE could overcome resistance to androgen receptor therapy [36]. This outcome further suggests a potential association between the AR and YAP1. The interaction between these two proteins holds biological significance because the increased expression of YAP1 promotes gene expression that is dependent on the AR, leading to the enhanced growth of PC cells in both in vitro and in vivo xenografts. However, further investigations are required to elucidate how alterations in YAP1’s functionality impact genome-wide interactions between the AR–DNA complex as well as transcriptional programs relevant to metastatic CRPC.

Autophagy is a cellular process involving the sequestration and subsequent degradation of intracellular components within lysosomes, contributing to cellular homeostasis by facilitating turnover and providing energy and macromolecular building blocks [8]. The role of autophagy in cancer is multifaceted and context dependent. Although some controversy exists, targeting autophagy has been proposed as a potential therapeutic strategy for cancer treatment [37,38]. Rapamycin, an FDA-approved drug known to activate autophagy via mTORC1, has demonstrated antitumor effects across various malignancies. The compound 3-methyladenine (3-MA) is commonly utilized as an autophagy suppressor owing to its capacity to inhibit the activity of class III PI3K, a pivotal factor in initiating autophagy. In our study, we observed that the rapamycin-mediated inhibition of YAP1 resulted in decreased expression levels of the AR and the PSA. Furthermore, 3-MA and the knockdown of ATG5 also activated the YAP1–AR–PSA axis (Figure 3).

Despite the notable success achieved through the immune checkpoint blockade in treating melanoma, there is still an inadequate response observed in “cold” tumors, like PC [39,40,41]. Given the promising potential of immunotherapy for PC treatment, we have gained further insights into specific targets associated with immune responses. Our findings demonstrate robust positive correlations between autophagy-related genes LC3B YAP1 and the AR and CD4+/8+ T cells in PC. Moreover, we find it particularly intriguing that high expression levels of YAP1 show significant associations with both T cells and APCs. This discovery has motivated us to delve deeper into exploring immune-related inflammatory factors, which have also piqued our interest. These observations imply that combining immunosuppressive agents synergistically could enhance the effectiveness of PC therapy (Figure 4 and Figure 5).

In conclusion, the findings of our study demonstrate the regulatory role of YAP1 in modulating PSA expression through the AR in CRPC cell lines, with autophagy activity influencing this process. Furthermore, our discoveries underscore the interconnectedness among autophagy, YAP1, and the AR in relation to immune T-cell infiltration and the intrinsic relationship between YAP1 and inflammatory factors, offering valuable insights into the development of innovative immune-based therapeutic strategies for CRPC.

## Figures and Tables

**Figure 1 biomedicines-12-00661-f001:**
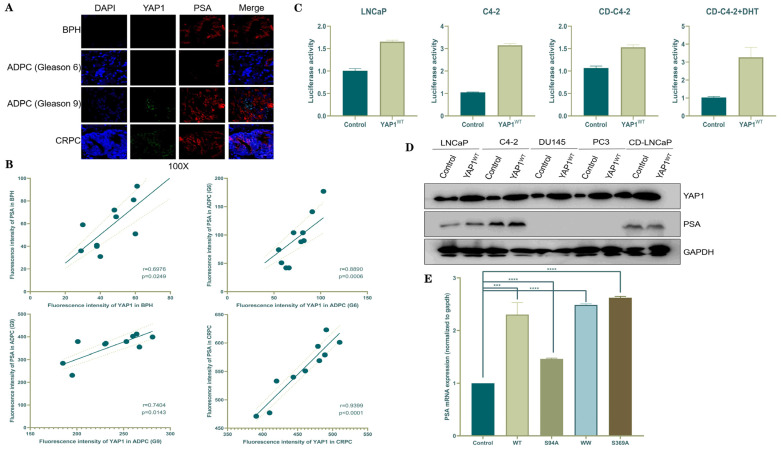
YAP1 and the PSA were colocalized and positively correlated in CRPC tissue and its cell lines. (**A**) IF assay analysis of the correlations between YAP1 and the PSA in different PC tissues (BPH, ADPC, and CRPC). (**B**) Correlation curves of YAP1 and PSA expressions in different PC tissues (BPH, ADPC, and CRPC) by quantification of immunofluorescence; ten sites are selected for each tissue. (**C**) Promega luciferase showed PSA gene expression after YAP1 overexpression in LNCaP and C4-2 cells, and the PSA gene expression after YAP1 overexpression in C4-2 cells treated with DHT. (**D**) WB experiments showed the changes in PSA expression after YAP1 overexpression in different PC cell lines; (**E**) qPCR detected the change in the PSA expression in C4-2 cells with YAP1 overexpression. *** *p* < 0.001; **** *p* < 0.0001.

**Figure 2 biomedicines-12-00661-f002:**
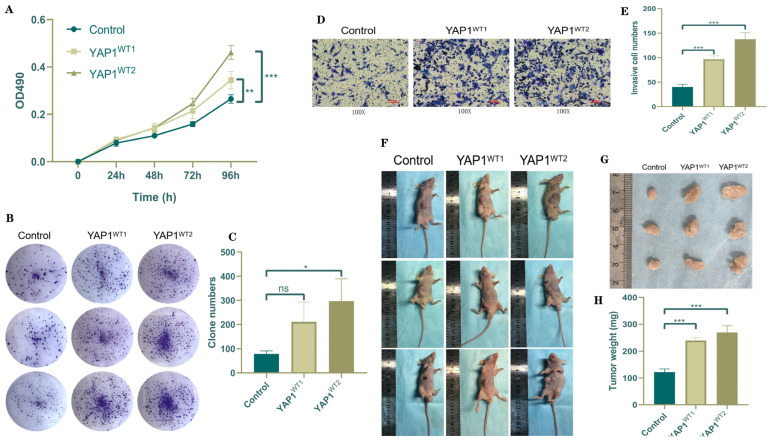
YAP1 promoted CRPC cell line proliferation and migration in vivo and in vitro. (**A**) MTT assays of C4-2 cells transfected with negative control and overexpression plasmid of YAP1. (**B**) Colony formation assays of C4-2 cells transfected with negative control and overexpression plasmid of YAP1. (**C**) Quantification of B. (**D**) Transwell assay of C4-2 cells transfected with negative control and overexpression plasmid of YAP1. (**E**) Quantification of D. (**F**,**G**) C4-2 cells transfected with negative control and overexpression plasmid of YAP1 were transplanted subcutaneously in nude mice *(n* = 3/per group). The effects of the negative control and overexpression plasmid of YAP1 on the growth of the PC. (**H**) The tumor weight was measured with a caliper. * *p* < 0.05; ** *p* < 0.01; *** *p* < 0.001; *p* > 0.05, ns.

**Figure 3 biomedicines-12-00661-f003:**
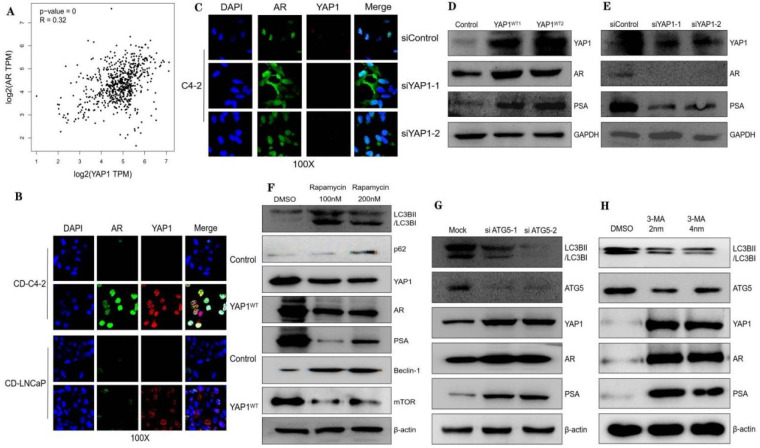
Autophagy regulates the YAP1/AR/PSA axis in the CRPC cell line. (**A**) The correlation between YAP1 and the AR was analyzed online (*p* < 0.05). (**B**) IF assay analysis of the correlation between YAP1 overexpression and the AR in C4-2 and LNCaP cells. (**C**) IF assay analysis of the correlation between YAP1 silencing and the AR in C4-2 cells. (**D**) Western blotting for detecting the expressions of the AR and PSA in C4-2 cells transfected with YAP1-overexpression plasmid. (**E**) Western blotting for detecting the expressions of the AR and PSA in C4-2 cells transfected with YAP1 siRNA. (**F**) Western blotting for detecting the expressions of YAP1, the AR, and PSA in C4-2 cells transfected with rapamycin. (**G**) Western blotting for detecting the expressions of YAP1, the AR, and PSA in C4-2 cells transfected with ATG5 siRNA. (**H**) Western blotting for detecting the expressions of YAP1, the AR, and PSA in C4-2 cells transfected with 3-MA.

**Figure 4 biomedicines-12-00661-f004:**
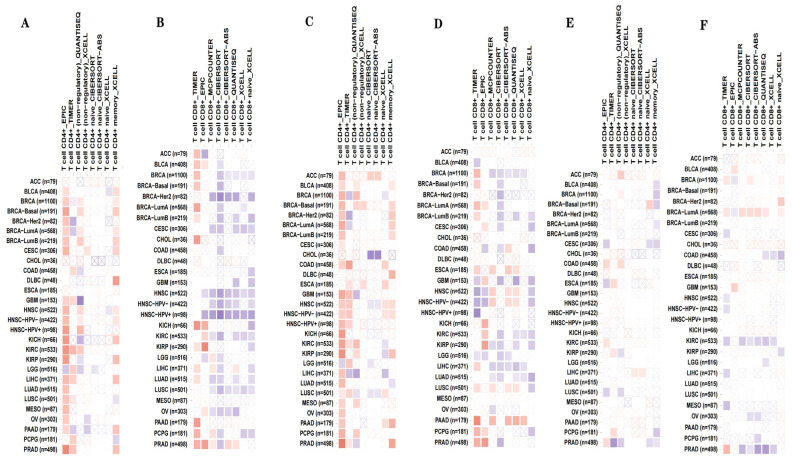
TIMER2.0 evaluates associations between immune infiltrates and YAP1, the AR, and PSA. (**A**) The association between YAP1 alterations and T-cell CD4+ immune infiltrates in different tumors. (**B**) The association between YAP1 alterations and T-cell CD8+ immune infiltrates in different tumors. (**C**) The association between AR alterations and T-cell CD4+ immune infiltrates in different tumors. (**D**) The association between AR alterations and T-cell CD8+ immune infiltrates in different tumors. (**E**) The association between KLK3 alterations and T-cell CD4+ immune infiltrates in different tumors. (**F**) The association between KLK3 alterations and T-cell CD8+ immune infiltrates in different tumors.

**Figure 5 biomedicines-12-00661-f005:**
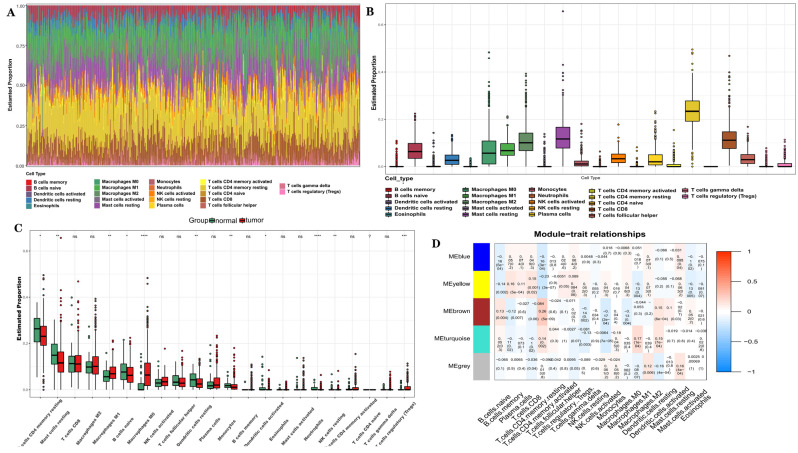
Immunoinfiltration in the combined sample dataset. (**A**) The bar chart shows the compositions of the 22 immunoinfiltrating cells in various samples, with each column representing a sample. (**B**) The histogram shows the compositions of the immunoinfiltrating cells in all the samples. (**C**) Comparison of the contents of various immunoinfiltrating cells in normal samples and PC samples. (**D**) Correlation heatmap between modules and immunoinfiltrating cells. * *p* < 0.05; ** *p* < 0.01; *** *p* < 0.001; **** *p* < 0.0001; *p* > 0.05, ns.

**Figure 6 biomedicines-12-00661-f006:**
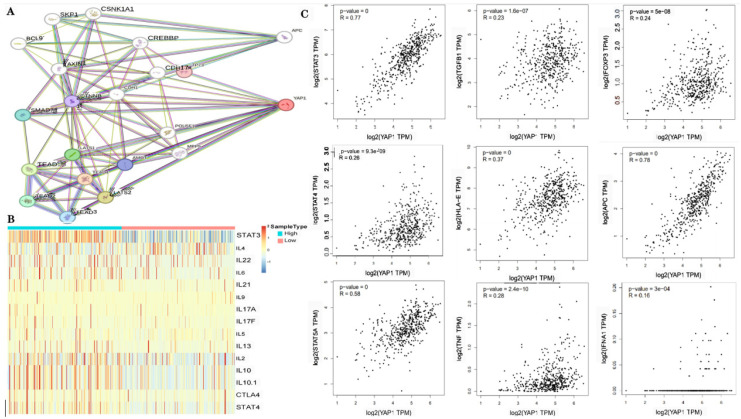
Correlations of YAP1 with immune stimulators and inflammatory cytokines. (**A**) Protein–protein interactions (PPIs) of YAP1 and antigen-presenting cell (APC). (**B**) Heatmap shows the relationships between different YAP1 groups (high/low) and immune stimulators/inflammatory factors in PC. (**C**) The correlations between YAP1 and immune stimulators/inflammatory factors were analyzed online.

## Data Availability

The data of the materials and methods and results to support the conclusions are included in this article. If any other data are needed, please contact the corresponding author.

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
