# Peer review of "YAP1 Regulates the YAP1/AR/PSA Axis through Autophagy in Castration-Resistant Prostate Cancer and Mediates T-Cell Immune and Inflammatory Cytokine Infiltration"

_biomedicines, 2024, doi:10.3390/biomedicines12030661_

Round 1
Reviewer 1 Report
Comments and Suggestions for Authors
The study focuses on YAP1 and its role in the regulation of the YAP1/AR/PSA axis through autophagy in castration-resistant prostate cancer with signs of T cell immune and inflammatory cytokines infiltration. The study although interesting, could provide more mechanistic insights. There are some main issues that need to be addressed:
1. Please make clear what is novel in this study.
2. In Figure 2B the purple color is too dark and the colonies are not visible.
3. Regarding the association with T cells (Fig 4 and 5) the results could be better presented, the authors should better clarify ithe cells involved and their role.
In addition, the point raised to employ immunosuppressive agents as a strategy does not make sense (line 381), taken into consideration the results.
4. There are no clear results to claim the this study uncovered a novel molecular mechanism..... (line 343-344), please avoid
5. In all figures the labeling-text in each axis is two small, hardly readable. Please keep a bigger text in all figures.
6. There are recent references that are not cited in the manuscript. Please update the literature.
Comments on the Quality of English Language
Moderate editing of English language is required.
Author Response
Dear reviewer,
I am very glad that my manuscript has been approved by you and thank you for pointing out the shortcomings of my manuscript. I have revised the original text according to your instructions. All the changes have been marked in red. I hope the revised manuscript can get your approval.
- I have given a more specific description of the novelty of the article in the manuscript, mainly in the discussion and conclusion sections. Thank you for the suggestion and this change has helped to make the text more fluid and readable.
- The clone formation experiment has been done again, thank you very much for your correction.
- The results associated with T cells (Fig. 4 and 5) have been illustrated in more detail. The roles of other cells, including applications in immunity, inflammation, autophagy, etc., are discussed in detail. We modified throughout the text according to the comment. Thank you very much.
- Thank you for underlining this deficiency. This section was removed according to the information showed in the work suggested by the reviewer.
- I have modified the labeling-text in figures according to your requirements.
- The references have been revised in accordance with your specifications. We greatly appreciate your invaluable feedback.
The manuscript has been embellished through WILEY EDITING SERVICES. Once again, thank you very much for your comments and suggestions.
Reviewer 2 Report
Comments and Suggestions for Authors
Perfect clinical study with clear aim, excellent results , and inclusion of an animal model. I have only one question.
Minor point:
Regarding autophagy, are there data on ferroptosis and cuproptosis being involved in this proocess?
Comments on the Quality of English LanguageAlmost perfect English
Author Response
Dear reviewer,
First of all, I would like to express my sincere thanks to you for taking time out of your busy schedule to help me review my manuscript. In response to your question, I hereby give you a detailed explanation.
There have been several articles on the link between YAP1 and ferroptosis. However, the relationship between YAP1 and ferroptosis is ambiguous, with some scholars suggesting that YAP1 promotes ferroptosis and others suggesting that it inhibits ferroptosis. In prostate cancer studies, YAP1 mainly promotes ferroptosis. At present, little is known about the association between cuproptosis and YAP1.
Finally, our research group would like to express our sincere thanks to you again.
Reviewer 3 Report
Comments and Suggestions for Authors
Please answer the comments of the reviewer one by one, point-by-point. Otherwise, it will be difficult to proceed with the review.
1. This study is rejected because it was not conducted as described in the title.
2. The above reason is to prove there is a correlation through the mechanism of autophagy. It is necessary to prove that various autophagy-related biomarkers (m-TOR, Beclin-1, LC3, p62, ATGs, etc.). Additionally, inhibitors (3-MA, Raf, and Baf) that are inhibited or activated can be applied. Based on the results, predict the optimal biomarkers using network pharmacology techniques or PCA analysis compared with the data in vitro, in vivo, and in silico.
3. The picture or image is too low in resolution to proceed with further review.
4. Many typos are found in the 'Materials and Methods' section.
Many typos are found in the 'Materials and Methods' section.
Author Response
Dear reviewer,
First of all, I feel very honored that our manuscript can be viewed by professional reviewers like you, and you have provided valuable suggestions. In response to your requirements, we have made modifications. All the modified fonts are marked in red. We hope to get your approval.
- We are sorry that some parts of the meaning and title of our manuscript may not be clearly expressed. We have tried our best to correct this mistake, and we hope you can give us another chance to revise it.
- I have further verified the changes of autophagy related markers (m-TOR, Beclin-1, LC3, p62, etc.) according to your requirements. In addition, inhibitors (3-MA) are also used to further strengthen the proof of the study. Thank you very much for your valuable advice.
- We have optimized the clarity and size of the figures, and we hope to get your approval.
- We'll describe the materials and methods section in more detail, and this part has been embellished through WILEY EDITING SERVICES. Once again, thank you very much for your comments and suggestions.
Our team would like to express our sincere thanks to you again. Thank you very much for your help in improving the quality of our manuscripts.
Reviewer 4 Report
Comments and Suggestions for Authors
Abstract:
Line 22, ADPC, HPH mention the full names when the abbreviations are appearing for the first time in the text.
The authors should rewrite the abstract mentioning briefly the experimental work and highlighting the results.
Introduction:
The aim of the work should be presented clearly at the introduction.
Line 42, “Patients with CRPC experience unfavorable clinical outcomes”. Please give examples for these outcomes.
Line 54 to 56, “Furthermore, compared to primary tumors, castration-resistant prostate tumors exhibit upregulated levels with highly activated YAP in our study”. Please rephrase this sentence, and avoid introducing the results in the introduction.
Line 72 to 75, “In the present study, we found that YAP1 was significantly upregulated in CRPC samples. Overexpression of autophagy induced AR-mediated PSA activation metastatic phenotype in androgen-independent PC cells by stimulating YAP activity. And YAP1-mediated YAP1/AR/PSA axis may rolled in CD4+ T Cell immune infiltration in PC”. This part is more relevant to the result section.
Material and method:
The authors should add updated references to material and method part.
The authors should add the ethical approval for recruiting patients and running the experimental setup with animals.
The authors should provide details of the used kits (catalogue number and vendors) .
Line 138, “MTT solution”. Add the full name (3-(4,5-dimethylthiazol-2-yl)-2,5-diphenyl tetrazolium bromide) to MTT abbreviation.
Line 156 to 161, please mention the mice models, weight, age, housing, feeding, type of collected data, specific time of collected data, and how the authors sacrificed the animals.
Results:
The authors should add the expression of data to figure legends, i.e columns carries ***……..refer to…at (p<0.001).
The figures are overcrowded. The authors should simplify and clarify the contents.
Line 138, “luciferase activity”, please mention the enzyme action in more details.
Line 174: it would be CRPC tissue not CPPC tissue.
Line 222 to 225, “Nude mice (~5 weeks old) were transfected with either YAP1 control plasmids or YAP1 WT plasmids and implanted subcutaneously under their abdomen skin. Following a period of 2 weeks, sterile conditions were maintained on an animal laboratory workbench for sacrificing the nude mice”. This part could be suitable for the material and method section.
Line 238, GEPIA online database, please describe in more details.
Line 255 to 261, this part could be transferred to discussion with citation.
Line 278 to 282, this part could be transferred also to discussion with citation.
Line 282, “Using TIMER2.0 software analysis”, please cite the software.
Line 286, “we utilized the CIBERSORT algorithm”, please cite the algorithm.
Line 289 to 294, this part could be transferred to discussion part.
Line 294 to 297, “Before constructing the co-expression matrix with weighted values, we set a soft-threshold β value of 6 to ensure a scale-free topology for analysis, achieving an independence degree of 0.8”. This part could be transferred to material and method part.
Line 297 to 311, rewrite this part highlighting the results only and transfer the rest to discussion part.
Discussion:
Authors should rewrite the discussion part again and compare their results with previous studies and highlight the new findings and their significance.
Conclusion:
This section needs to be rewritten where the future perspective is added.
The authors should add graphic abstract.
Comments on the Quality of English Language
English editing is highly recommended.
Author Response
Dear reviewer,
I am very glad that my manuscript has been approved by you and thank you for pointing out the shortcomings of my manuscript. I have revised the original text according to your instructions. I hope the revised manuscript can get your approval. The changes have been marked in red.
Abstract
- Thank you for underlining this deficiency. We have described the full names of ADPC and BPH.
- We have described the summary in more detail. We cut out unnecessary parts and highlight the results of our work. Thank you very much for your valuable advice.
Introduction
- In the introduction section we redescribe the purpose of the study. Thank you very much.
- Line 42, “Patients with CRPC experience unfavorable clinical outcomes”. We have included specific unfavorable clinical outcomes of CRPC patients and references as per your guidance to prove.
- Line 54 to 56, “Furthermore, compared to primary tumors, castration-resistant prostate tumors exhibit upregulated levels with highly activated YAP in our study”. We have rephrased that sentence, thank you very much.
- Line 72 to 75, “In the present study, we found that YAP1 was significantly upregulated in CRPC samples. Overexpression of autophagy induced AR-mediated PSA activation metastatic phenotype in androgen-independent PC cells by stimulating YAP activity. And YAP1-mediated YAP1/AR/PSA axis may rolled in CD4+ T Cell immune infiltration in PC”. We have deleted and corrected this part. Thank you for your valuable advice.
Material and method
- We have added new references in the Materials and Methods section.
- Ethical approval details including Ethics number have been sent to the editor. Thank you very much for your comments.
- According to your request, the full name of MTT is described in the article (Line 138). Thank you very much.
- We are following your request. mentioned the mice models, weight, age, housing, feeding, type of collected data, specific time of collected data, and how the authors sacrificed the animals (Line 156-161). Thank you for your valuable advice.
Result
- We have added the expression of data to figure legends according to your request. Thank you very much for your guidance.
- We have simplified and clarified the contents of figures.
- We describe the role of enzymes in more detail (Line 138).
- We are very sorry for this mistake, and we have corrected the word CPPC (Line 174).
- Thanks for your patience and correction, we have removed this part to the Materials and methods section (Line 222-225).
- We describe GEPIA, an online database, in more detail. Once again, we express our sincere gratitude (Line 238).
- We've moved this part to the discussion section (Line 255-261).
- We've moved this part to the discussion section (Line 278-282).
- We have cited TIMER2.0 software (Line 282).
- We have cited CIBERSORT algorithm (Line 286).
- We've moved this part to the discussion section (Line 289-294).
- Thanks for your patience and correction, we have removed this part to the Materials and methods section (Line 294-297).
- We have revised this section and put the rest of the content in the discussion section as you requested (Line 297-311). Thank you very much.
Discussion
- According to the above changes, we have made new changes to the discussion section and highlighted the focus of our research.
Conclusion
- We have re-described this section and added our future perspective. Thank you very much for your valuable advice.
We have uploaded a graphic abstract and refined the article through the WILEY EDITING SERVICES. I would like to express my sincere blessing to you.
Reviewer 5 Report
Comments and Suggestions for Authors
Dear Authors,
The paper titled "YAP1 regulates the YAP1/AR/PSA axis through autophagy in castration-resistant prostate cancer and mediates T cell immune and inflammatory cytokines infiltration" presents intriguing findings and introduces innovative elements. However, I would like to offer some feedback and highlight both the strengths and weaknesses of the manuscript:
Detailed Comments:
1. The Abstract lacks structure. The research objective should be presented immediately after the introductory sentence.
2. The Introduction inadequately introduces the reader to the research subject. Additionally, a clear research objective or an alternative research hypothesis should be presented at the end of the introduction, rather than a conclusion from the research.
3. The "Materials and Methods" chapter is well-organized with subsections, enhancing readability.
4. The "Results" chapter needs further elaboration.
5. The "Discussion" section could benefit from more in-depth analysis.
Strengths:
1. Strong evidence for the role of YAP1 in CRPC: The study provides compelling evidence for the significant role of YAP1 in the progression of castration-resistant prostate cancer (CRPC). Identified mechanisms, such as the activation of the YAP1/AR/PSA pathway and its impact on AR nuclear translocation, are well-described.
2. Connection with autophagy: The study focuses on the link between YAP1 and autophagy, crucial for understanding the molecular mechanisms influencing CRPC progression. The use of rapamycin as an autophagy activator and the analysis of its impact on AR and PSA expression add depth to the experiments.
3. Association with the immune system: The study identifies the connection between YAP1, AR, and T cells of the immune system. Suggestions regarding potential targets for immunotherapy against CRPC are essential, especially in the context of contemporary cancer therapies.
4. Analysis of protein interactions: Presenting YAP1 protein interactions with antigen-presenting cells (APC) and correlating them with immunological factors adds complexity to understanding the role of YAP1 in the immunological context.
Weaknesses:
1. Lack of clinical studies: The research relies mainly on cellular and animal analysis, lacking direct clinical studies on humans. Further clinical studies would be necessary to assess the impact of the results on clinical practice.
2. Need for additional evidence of clinical relevance: Although the study suggests a role for YAP1 in CRPC, further research is needed to confirm these results in more complex clinical conditions. Studies with a larger number of patients are recommended.
3. Impact on other research areas: The study primarily focuses on the role of YAP1 in the context of CRPC and its interaction with the immune system. However, the impact of YAP1 on other areas of cell biology and signaling requires additional investigation.
4. Lack of analysis of potential side effects: When examining the impact of rapamycin on YAP1 and autophagy, a detailed analysis of potential side effects of this drug is missing, which is crucial in the context of potential clinical application.
Suggestions for Improvement:
1. Expand clinical studies: Inclusion of clinical studies in humans would directly translate the findings into clinical practice. Consideration of various clinical cases and assessment of the therapy's impact on patient outcomes is advisable.
2. Long-term analysis of drug effects: Regarding the use of rapamycin as an autophagy activator, it is essential to investigate the long-term effects of this drug, including potential side effects. This will aid in evaluating its clinical applicability.
3. Increase complexity of studied processes: The study could be more comprehensive by considering other aspects of cell biology and signaling that may influence CRPC development. This will contribute to a fuller understanding of the mechanisms involved in the disease.
4. More patients in immunological analysis: To confirm the associations between YAP1 and the immune system, an increase in the number of patients in immunological analyses will provide more representative results.
5. Provide more information on methodology: Additional information on the research methodology, especially regarding cell analyses, will enhance transparency and enable other researchers to replicate the experiments.
6. Detailed analysis of autophagy results: A more in-depth analysis of autophagy-related results, especially concerning the prolonged use of rapamycin, would provide more information about potential autophagy-dependent mechanisms in the studied processes.
7.Emphasize practical implications: Clearly highlighting the practical implications of the findings, especially in the context of potential therapeutic targets, will increase the significance of the study for the medical community.

Comments on the Quality of English LanguageMinor English editing is required.
Author Response
Dear reviewer,
I am very glad that my manuscript has been approved by you and thank you for pointing out the shortcomings of my manuscript. I have revised the original text according to your instructions. All the changes have been marked in red. I hope the revised manuscript can get your approval.
Detailed Comments
- We have made the abstract more structured and put the excess part in the introduction section. Thank you very much for your correction.
- We have made detailed modifications to the introduction, including specific descriptions of target proteins and autophagy. In addition, we have modified the research conclusions in the introduction to the research hypothesis.
- Thank you very much for your satisfaction with our section and thank you for your patient guidance.
- We have elaborated the results in more detail to make our results more convincing.
- We also explained the discussion part in more detail to make the conclusion more readable. Your suggestions have greatly improved the quality of our article.
Strengths
We really appreciate your recognition of our research. We are very happy that our research content can be liked and accepted by others. Your every word has increased our research enthusiasm.
Weaknesses
Thank you very much for pointing out the shortcomings of my research. We will try our best to revise our manuscript to meet your requirements. We also hope that you can give me the opportunity to revise it.
Suggestions for Improvement
- According to your request, we collected dozens of cases of CRPC patients' clinical information, including age, family history, treatment, operation, recurrence, metastasis and other detailed information. This part has been uploaded to supplement table 1 and we hope to get your approval. We're also working with clinicians to better evaluate medications to see if they can be used in the clinic.
- Regarding the impact of rapamycin on treatment, as well as its specific role and control of medication duration, we have described it more carefully through relevant clinical guidance and the latest literature. Thank you for your kind advice.
- We have elaborated on the cell biology and signaling of CRPC in more detail, focusing on autophagy and YAP1 molecules in general. Thank you for your guidance. We appreciate it.
- We added more patient analysis to the immune module to make the data more persuasive. Thank you for your reminding.
- We have explained the methods section in more detail and thank you for your valuable comments.
- We conducted a more in-depth analysis of the impact of autophagy in the results, including a summary of the latest literature and our own understanding of autophagy.
- We have carried out detailed statistics on the practical significance of this study. At the same time, I also find out similar references from other people's literature for comparison, hoping to make readers have a better understanding of this paper.
Our team would like to express our sincere thanks to you again. Thank you very much for your help in improving the quality of our manuscripts.
Round 2
Reviewer 1 Report
Comments and Suggestions for Authors
There are no Figures in the submitted file, please forward the figures
Author Response
Dear review,
I'm sorry for causing you this misunderstanding. We have uploaded all the revised figures, including supplementary materials, but these figures are submitted separately from the revised manuscript. I am deeply ashamed of the inconvenience I have caused you, and I will upload a revised manuscript with altered figures. Sincere wishes to you again.
Yours,
Ning Jiang
Reviewer 3 Report
Comments and Suggestions for Authors
Thank you for submitting to Biomediciens and for the revisions. My feedback is as follows:
-
1. The plagiarism check of the revised manuscript conducted by the editorial office yielded a result of 38%. I was particularly surprised to find numerous sentences in the Discussion section that closely resemble previously published work.
-
2. There is a significant disparity in the intensity of the control bands in Figure 1 of the western blotting. It appears that repeat experiments are necessary, and a positive control is missing.
-
3. In all experiments, it is essential to include a positive control drug to compare the expression of autophagy mediated by YAP1 and PSA.
-
The resolution of Figures 4 to 5 is insufficient, making it challenging to provide feedback for the review process.
4. Please address these points in your revisions to enhance the quality and integrity of the manuscript. Strongly recommend to submit other journals. Thank you.
Comments on the Quality of English LanguageThis is lacking in originality.
Author Response
Dear reviewer,
We have carried out professional weight reduction on the manuscript, and now the manuscript repetition rate is 18%, which meets the publication requirements of the magazine. In addition, we have re-adjusted the sharpness of figure 4 and 5. Thank you for your correction.
Yours,
Ning Jiang
Reviewer 4 Report
Comments and Suggestions for Authors
Abstract
- Move the sentence “This study aims to investigate potential … with autophagy” to follow the sentence “(ADT) is associated with …….. limited treatment options.”
- “Additionally, we observed…………………………on PSA expression within CRPC.”, remove “we” and rephrase.
Material and Methods Section:
- Please provide updated references to support this section.
- Figures 4, 5, and 6 are currently missing. Please ensure they are included in the manuscript.
Comments on the Quality of English LanguageAuthors, correct any typographical errors.
Author Response
Dear reviewer,
Once again, I would like to express my sincere thanks to you for your correction of the shortcomings of our manuscript.
Abstract
- We have moved the sentence “This study aims to investigate potential … with autophagy” to follow the sentence “(ADT) is associated with …… .. limited treatment options.” Thank you very much for your patient guidance.
- We have removed "we" and rephrased this sentence. Thank you for your valuable advice.
Material and Methods
- We would like to thank you for supplementing the references in this section.
- All images in the last revised version have been uploaded in a separate file and are separate from the revised manuscript. I would like to apologize to you for the bad review experience caused by my negligence. I was so ashamed. I have uploaded the revised manuscript including all altered figures.
I would like to express my best wishes to you again.
Yours,
Ning Jiang
Reviewer 5 Report
Comments and Suggestions for Authors
Dear Authors,
Review of the work entitled “YAP1 regulates the YAP1/AR/PSA axis through autophagyin castration-resistant prostate cancer and mediates T cell immunity and inflammatory cytokines infiltration.”
Detailed review of the revised paper:
1.
Abstract: The abstract provides essential information about the study, including background, objectives, findings, and conclusions. The language is precise and technical, typical of scientific work. The research problem is clearly formulated. Key information about the research methodology is provided. The summary of the study's conclusions is clear.
2.
Introduction: The introduction provides context for the study by presenting information about the problem of CRPC and relevant epidemiological data. It also includes information about the significance of the Hippo-YAP pathway in PC. Reference to previous studies allows for a continuation of the discussion on the role of YAP1 in PC.
3.
Materials and Methods: The materials and methods are described in detail, enabling other scientists to replicate the experiments. The description of laboratory and experimental procedures is exhaustive. Statistical analysis: The statistical analysis is conducted in accordance with applicable standards. The statistical tests used are appropriately chosen for the nature of the data.
4.
Results: The research results are presented in a clear and logical manner. The use of various techniques, such as Western blot or immunofluorescence, to analyze the expression of proteins and the localization of YAP1, AR, and PSA, allows for a comprehensive understanding of the phenomena studied in the work.
5.
Discussion: The discussion interprets the research results in the context of the scientific literature. The authors emphasize the significance of their findings and suggest potential clinical implications.
6.
The summary of the paper emphasizes its significance and suggests further research directions.
7.
Conclusions: The conclusions are logical and well-supported by the gathered scientific evidence. The conclusions align with the findings presented in the text.
Overall assessment: The reviewed paper meets the requirements of scientific work. It is well-organized, contains comprehensive information, and logically guides the reader through all stages of the study. The language is technical and scientific, which is appropriate for this type of publication. Overall, the paper is a solid scientific article that contributes significantly to understanding the pathophysiological mechanisms associated with CRPC and potential therapies.

Author Response
Dear reviewer,
Each of your careful comments on the manuscript moved us very much and made us more confident about the content of our own research. Thank you very much for your recognition of our manuscript. We feel very honored to have our manuscript reviewed by such a professional and responsible reviewer as you. Once again, our best wishes and thanks to you.
Yours,
Ning Jiang